# Development and Validation of Three Triplex Real-Time RT-PCR Assays for Typing African Horse Sickness Virus: Utility for Disease Control and Other Laboratory Applications

**DOI:** 10.3390/v16030470

**Published:** 2024-03-20

**Authors:** Rubén Villalba, Cristina Tena-Tomás, María José Ruano, Marta Valero-Lorenzo, Ana López-Herranz, Cristina Cano-Gómez, Montserrat Agüero

**Affiliations:** 1Laboratorio Central de Veterinaria, Ministry of Agriculture, Fisheries and Food, 28110 Algete, Spain; rvillalba@mapa.es (R.V.); mruanor@mapa.es (M.J.R.); mvalero@mapa.es (M.V.-L.); alherranz@mapa.es (A.L.-H.); ccano@mapa.es (C.C.-G.); 2Tecnologías y Servicios Agrarios, S.A. (TRAGSATEC), 28037 Madrid, Spain; at_algete9@mapa.es

**Keywords:** African horse sickness, real-time RT-PCR, serotypes, diagnostic, vaccination

## Abstract

The African horse sickness virus (AHSV) belongs to the Genus Orbivirus, family Sedoreoviridae, and nine serotypes of the virus have been described to date. The AHSV genome is composed of ten linear segments of double-stranded (ds) RNA, numbered in decreasing size order (Seg-1 to Seg-10). Genome segment 2 (Seg-2) encodes outer-capsid protein VP2, the most variable AHSV protein and the primary target for neutralizing antibodies. Consequently, Seg-2 determines the identity of the virus serotype. An African horse sickness (AHS) outbreak in an AHS-free status country requires identifying the serotype as soon as possible to implement a serotype-specific vaccination program. Considering that nowadays ‘polyvalent live attenuated’ is the only commercially available vaccination strategy to control the disease, field and vaccine strains of different serotypes could co-circulate. Additionally, in AHS-endemic countries, more than one serotype is often circulating at the same time. Therefore, a strategy to rapidly determine the virus serotype in an AHS-positive sample is strongly recommended in both epidemiological situations. The main objective of this study is to describe the development and validation of three triplex real-time RT-PCR (rRT-PCR) methods for rapid AHSV serotype detection. Samples from recent AHS outbreaks in Kenia (2015–2017), Thailand (2020), and Nigeria (2023), and from the AHS outbreak in Spain (1987–1990), were included in the study for the validation of these methods.

## 1. Introduction

AHS is an infectious, non-contagious viral disease affecting all species of *equidae*, caused by an *Orbivirus* of the *Sedoreoviridae* family, transmitted by different *Culicoides* biting midge species. Nine different serotypes (1–9) of AHSV have been identified. The disease can evolve into four clinical presentations (horse sickness fever, cardiac form, mixed form, and pulmonary form) with different degrees of severity that, in the case of the respiratory form, can exceed 95% of mortality in susceptible non-vaccinated animals [1,2].

AHS is endemic in several countries of Sub-Saharan Africa. Outside endemic areas, outbreaks produced by AHSV-9 were reported in 1959–1961 in Saudi Arabia, Lebanon, Syria, Jordan, Iraq, Turkey, Cyprus, Iran, Afghanistan, Pakistan, and India. In 1965, AHSV-9 outbreaks were recorded in Morocco, Algeria, and Tunisia and, one year later, the disease appeared in Spain. In September 1987, a new outbreak of AHS caused by serotype 4 was reported in Spain, and in Morocco and Portugal in the following years [2,3]. Recently, in early 2020, an unexpected outbreak of AHS produced by a serotype 1 strain appeared in Thailand [4,5], affecting Malaysia too [6] and threatening the equine industry worldwide.

The prevention and control of AHS is an international priority in animal health. AHS is a notifiable animal disease by the World Organisation for Animal Health (WOAH) [7] and one of the seven animal diseases currently included by the WOAH in the procedure for official recognition of a country’s disease-free status [8]. In addition, the European Union’s (EU) “Animal Health Law” [9] includes AHS as one of the five diseases specifically named for immediate eradication if detected within the EU.

The fight against the disease is based on controlling equid movement between free and infected zones [10] and systematic vaccination in endemic areas or in free areas in case of an outbreak. Currently, the only licensed vaccine in the world is a polyvalent live-attenuated vaccine (LAV), manufactured by Onderstepoort Biological Products (OBP), South Africa: African horse sickness vaccine for horses, mules and donkeys (Reg. No. G 0116 (Act 36/1947) Namibia: NSR 0586). Since 1994, this vaccine has consisted of two vials. The first vial contains a mixture of serotypes 1, 3, and 4, and the second vial contains serotypes 2, 6, 7, and 8. Serotypes 5 and 9 are not included in the vaccine because of cross-protection with serotypes 8 and 6, respectively, which is well-known [11,12].

AHSV comprises non-enveloped viral particles consisting of a ten-segmented double-stranded (ds) RNA genome (Seg-1–10) encapsulated in the VP3 subcore, and surrounded by the serogroup (or species) specific protein VP7 (core). The outer shell of the particle is formed by the serotype-specific proteins VP2 and VP5 [13]. The serotype is defined by the interaction of neutralizing antibodies mainly with the VP2 outer capsid protein, as it is in the case of the Orbivirus type species, bluetongue virus (BTV). Nine different serotypes (named 1 to 9) of AHSV have been identified. Given that VP2 exhibits high levels of amino acid sequence variability between serotypes, it is not surprising that low levels of cross-protection between serotypes exist, which complicates vaccination strategies [11]. Additionally, the fact that orbiviruses have a segmented genome allows them to incorporate segments picked up from any “parental” virus when two orbiviruses of the same species co-infect the same cell. This process is called “reassortment” and represents a way for segmented viruses to evolve and generate new strains with different phenotypic characteristics [14].

AHS clinical signs and histological lesions are not pathognomonic. Although differential characteristics of the presentation of the disease can direct a field diagnosis, only laboratory tests can provide a definitive diagnosis. The Laboratorio Central de Veterinaria (LCV), based in Algete (Madrid, Spain), is the European Union reference laboratory for AHS and BT (EU-RL) and a WOAH reference laboratory for AHS [15,16]. As the EU-RL, the LCV provides scientific and technical support to the EU national reference laboratories of EU member states and harmonizes laboratory diagnostic methods as a key element for a rapid response to disease alerts for AHS.

The first step in approaching the AHS diagnosis is serogroup-specific (GS) detection. Currently, GS rRT-PCR for AHSV genome detection in organs and EDTA blood samples, as well as GS ELISA for AHSV antibodies detection in serum samples, are the main diagnostic tools used in the surveillance and control of AHS [17,18]. Secondly, serotyping must be performed as soon as possible to determine an adequate vaccination strategy. Virus neutralisation tests (VNT) and seroneutralisation tests (SNT) are the classical methods for AHSV serotyping [17,18]. In fact, the nine serotypes of AHSV were established based on these techniques. However, these procedures are time-consuming, cross-reactive (between serotypes 1 and 2, serotypes 3 and 7, serotypes 5 and 8, and serotypes 6 and 9) [11,12], often associated with poor sensitivity, and require having the virus isolated in cell culture.

Serotype-specific (TS) molecular tests targeted to the AHSV genome Seg-2 have been developed since 2000. These include probe-hybridisation methods [19,20], gel-based RT-PCR [21,22], and real-time RT-PCR (rRT-PCR) [23,24,25], which are based on the detection of Seg-2 and have already been described. However, considering the relevance of this disease and the high variability of seg-2 even among virus strains of the same serotype, it would be highly advisable to have additional rRT-PCR typing methods that would provide more versatility to the diagnosis.

This study describes the development, optimisation, and validation of a set of three triplex rRT-PCR assays which provide rapid and reliable AHSV-serotype identification and are suitable for high throughput diagnostic systems. Serotypes 2, 4, and 9 were included in the first triplex assay because, historically, outbreaks out of endemic areas involved those serotypes. Serotypes 3, 5, and 7 were included in the second triplex rRT-PCR, and serotypes 1, 6, and 8 in the third triplex rRT-PCR. They allow discrimination of mixed infections and have been used to identify incursions of multiple AHSV types in endemic and non-endemic situations.

## 2. Materials and Methods

### 2.1. Clinical Samples and Virus Isolates

A panel of AHSV isolates, including nine AHSV reference strains (serotypes 1 to 9) before and after cloning by end point dilution, AHSV live attenuated vaccine (vials 1 and 2) and their corresponding AHSV vaccine strains (serotypes 1–4 and 6–8) obtained after cloning by end point dilution, and a collection of 27 AHSV field isolates were used for the validation of rRT-PCR assays. Representative isolates of other orbiviruses, such as BTV reference strains for notifiable serotypes (serotypes 1 to 24), Epizootic hemorrhagic disease virus serotypes 1, 2, 4, 5, 6, 7, and 8, and Equine encephalosis virus (EEV) serotype 3, as well as viruses which produce diseases in equidae (West Nile virus linage 1 and Equine herpesvirus serotypes 1 and 4) were used to assess specificity of assays. All orbivirus isolates used in the study were obtained from the EU-RL collection and were grown in BHK-21 clone 13 cells (American Type Culture Collection, ATCC-CCL-10™), Vero cell monolayers (ATCC-CCL81), or in KC cells derived from *Culicoides sonorensis* [26]. Uninfected cell lines were analysed by evaluated methods. Clinical samples of equines were also studied, including the following: 30 EDTA blood samples from Spanish horses obtained in 2019–2022 (AHS-free status area), 62 blood and tissue samples from outbreaks in endemic countries, such as Kenia (2015–2017) and Nigeria (2023), and 98 blood and tissue samples from outbreaks in free countries, including 15 samples from Thailand (2020) and 83 samples from the last AHS outbreak in Spain (1987–1990), divided into 19 samples from the years 1987–1988 (polyvalent LAV was used for control purpose) and 64 samples from the years 1989–1990 (vaccination using monovalent AHSV-4 LAV was applied). Finally, to complete the assessment of diagnostic sensitivity and specificity, a panel of 50 clinical samples, including negative and positive samples of unknown serotypes from a ring trial performed in 2015 among WOAH reference laboratories for AHS [27], was analysed by TS rRT-PCR.

EDTA blood samples were stored between 2 and 8 °C until laboratory analysis. Tissue samples were obtained after necropsy and stored at deep-freezing temperatures (−80 °C). Viral suspensions were obtained after passages in cell culture (KC, Vero or BHK-21), clarified by centrifugation for 10 min at 2000 rpm, and stored at deep-freezing temperatures (−80 °C).

### 2.2. Virus Cloning by End Point Dilution of AHSV Suspensions Containing More than One AHSV Serotype (Polyvalent Commercial LAV and Original AHSV Reference Strains)

AHSV suspensions were ten-fold diluted (10^−1^ to 10^−6^) in EMEM. In cases of polyvalent commercial LAV, each vaccine suspension was previously incubated (*v*/*v* 50%) for 1 h at 37 °C with specific antisera against the serotypes included in the vaccine vial that were not intended to be isolated. Each dilution was added to 8 wells (50 µL/well) in 96-well-plates together with 50 µL of a Vero cells suspension containing 2 × 10^5^ cells/mL and incubated at 37 °C and 5% CO_2_ for seven days. Plates were observed daily under the optical microscope (4× and 10×). For each viral suspension, one well of the last dilution in which cytopathic effect (CPE) appeared was recovered to carry out a second round of cloning following the same procedure without specific antisera. Finally, one well was recovered to carry out TS rRT-PCRs to determine the AHSV serotype.

### 2.3. Nucleid Acid Extraction

Nucleic acid was extracted from 200 µL of EDTA blood, tissue homogenate or viral suspension using a BioSprint^®^ 96 DNA Blood Kit (Qiagen, Hilden, Germany) according to the manufacturer’s instructions. Nucleid acid was eluted in a final volume of 50 µL of nuclease-free water and kept at −80 °C until used.

### 2.4. Primers and Probes Design

Sequences encoding VP2 of each AHSV serotype available at the GenBank database were aligned using ClustalW 1.8. Target regions were identified for serotype-specific primers and probes, which were designed using Primer ExpressTM v2.0 (Applied Biosystems, Foster City, CA, USA). In silico studies were performed to ensure no cross-reaction with non-targeted sequences. Primers and probes sequences were previously published by Durán-Ferrer M. et al. in 2022 [28] (Table 1). Probes were labelled with compatible fluorophores to be combined in triplex assays for serotypes 2-4-9, 3-5-7, or 1-6-8.

### 2.5. Optimisation of rRT-PCRs in Triplex

To optimize rRT-PCRs in triplex, different concentrations of primers and probes and thermocycling profiles were tested. Finally, triplex TP rRT-PCR assays were performed in a final volume of 20 μL using the commercial kit AgPath-ID™ One-Step RT-PCR Reagents (Applied BioSystems, Whaltman, MA, USA). To summarize, 2 μL of RNA was mixed with 0.25 μM of each forward and reverse primer. The mix was denatured by heating at 95 °C for 5 min, followed by rapid cooling on ice. Then, 13 μL of RT-PCR reaction mix, which consisted of 10 μL of 2× RT-PCR buffer, 0.8 μL of 25× RT-PCR enzyme, and 0.0625 µM of each Taqman probe, was added to the primer mix. Reactions were performed using 7500 fast real-time PCR systems (Applied Biosystems) with the following program: 10 min at 48 °C and 10 min at 95 °C, followed by 40 cycles of 15 s at 95 °C, and 30 s at 55 °C. A non-template control (NTC) and positive controls for each serotype detected in the triplex (AHSV RNA) were also included. The rRT-PCR data were analysed with the StepOne software, version 2 (Applied Biosystems). The same conditions were used to perform rRT-PCR to detect a single serotype, in which case the volume of the primers and probes not included was substituted by RNase-free water.

### 2.6. Validation Parameters Evaluated

Analytical sensitivity of TS rRT-PCRs triplex and single assays was determined using serial ten-fold dilutions of the nine AHSV reference strains (after cloning) and compared to the results obtained using the GS rRT-PCR described by Agüero [29,30]. Intra-assay and intra-laboratory repeatability were evaluated for the nine serotypes in the triplex assays performing these tests in duplicate in two different assays (*n* = 4), and standard deviation was calculated for each sample. To estimate the correlation coefficient (R^2^) and efficiency (E) for each serotype in the triplex assays, standard curves for the nine serotypes were generated by linear regression on ten-fold serial dilutions (−1 to −7). Efficiency was defined as the fraction of target molecules that are copied in one PCR cycle calculated by the formula E (%) = (10^−1/slope^ − 1) × 100. Analytical specificity (exclusivity) was evaluated by analysing the uninfected cell suspensions usually used to propagate the AHSV strains (BHK-21, Vero and KC cells), heterologous orbiviruses (BTV reference strain serotypes 1–24, EHDV reference strain serotypes 1, 2, 4, 5, 6, 7, and 8, and Equine encephalosis virus serotype 3) and other viruses causing disease in Equidae (Equine herpesvirus serotypes 1 and 4; West Nile virus, strain Eg101). In addition, exclusivity among AHSV serotypes was tested using AHSV reference strains (before and after cloning), polyvalent commercial LAV (vials 1 and 2), and LAV strains after cloning. To evaluate the diagnostic sensitivity, the collection of AHSV field isolates and the panel of clinical samples from naturally infected horses positive or inconclusive by GS rRT-PCR, as well as a panel of 50 samples from the WOAH ring trial, were typed by the nine TS rRT-PCR in triplex assays. Diagnostic specificity was tested by analysing the panel of 30 EDTA blood samples from free-status countries by the nine TS rRT-PCRs in triplex, after confirming they were AHSV negative by the GS rRT-PCR.

### 2.7. Ethical Statement

The diagnostic samples collected from equids analysed in this study were taken from animals as part of veterinary investigations. Further ethical approval was, therefore, not obtained.

## 3. Results

### 3.1. Analytical Sensitivity

Nine AHSV reference strains (after cloning) were analysed in a ten-fold dilution (−1 to −7) by using each TS rRT-PCR targeted to segment 2, both in triplex and single, as well as the GS rRT-PCR targeted to segment 7 described in the WOAH Manual (Agüero, 2008). All samples were analysed in two different assays for each method, and in the case of TS rRT-PCR triplex and single assays, the samples were analysed in replicate in each assay (Appendix A). Triplex and single TS rRT-PCRs were similar or slightly more sensitive than GS rRT-PCR for the nine AHSV serotypes (Table 2). Correlation coefficients (R^2^) were greater than 98% for all serotypes, indicating a high predictive value in the regression lines. The qRT-PCR efficiencies for the triplex assays were higher than 80% for all serotypes, and wide dynamic ranges (at least six ten-fold dilutions) were reached for each serotype (Figure 1).

### 3.2. Analytical Specificity

Exclusivity

All non-AHSV orbiviruses analysed (BTV reference strain serotypes 1–24, EHDV reference strain serotypes 1, 2, 4, 5, 6, 7, and 8, and Equine encephalosis virus serotype 3) as well as other viruses that cause diseases in horses (Equine herpesvirus serotypes 1 and 4, and West Nile virus lineage 1) were negative in the triplex rRT-PCR assays. All uninfected cell cultures (BHK-21, Vero, and KC) analysed in the triplex assays were negative, except those grown in Vero cells when analysed in triplex 1-6-8 rRT-PCR, which produced a weak positive result (Ct 34.8) with an AHSV-8 probe. This non-specific result did not appear when the same samples were analysed in the TS rRT-PCR AHSV-8 single assay. This finding was observed in most virus suspensions produced in Vero cells evaluated in TS rRT-PCR triplex assay (Appendix A).

Exclusivity among serotypes

Six out of nine AHSV reference strains analysed before cloning by TS rRT-PCR triplex assays, showed a positive result for unexpected AHSV serotypes (Table 2). These unexpected results were confirmed by partial Sanger sequencing of segment 2, indicating that those viral suspensions contained mixed AHSV serotypes. To obtain a homogenous viral population for each serotype, a process of cloning by limiting dilution was performed on viral suspensions showing mixed serotypes as described above, and the purity of the cloned virus was confirmed by TS rRT-PCR triplex assays (Table 2). The same cloning process was performed on vials 1 and 2 of LAV which contained mixed AHSV serotypes in high concentration to obtain separate vaccine viruses of each serotype. The results of the analysis by TS rRT-PCR triplex assays before and after the cloning process confirmed that the cloned viral suspension contained a single serotype (Table 2).

### 3.3. Diagnostic Sensitivity

All field viral isolates of known serotypes were correctly typed by using triplex rRT-PCR (Appendix A). The expected serotype was found in all of them. Clinical samples were successfully typed by TS rRT-PCR triplex assays, even those with weak positive results in the GS rRT-PCR. The results obtained were compatible with the different epidemiological situations according to the source of the samples. Firstly, regarding the Spanish outbreaks, all 64 samples from 1989 to 1990 were typed exclusively as AHSV-4, while serotype 4 was identified together with other AHSV serotypes in some samples from 1987 to 1988 (specifically serotypes 2, 3, 5, and 9, as shown in Appendix A). Secondly, in 2020 Thailand’s outbreak serotype 1 was identified in all samples and serotype 3 was jointly detected in one of them (Appendix A). Thirdly, serotype 9 was detected in all samples from the Nigerian outbreak of 2023 (Appendix A). Finally, it was possible to establish the AHSV serotype in all untyped samples from Kenian outbreaks of 2015–2017 (Appendix A) and in 38 out of 39 GS rRT-PCR positive samples from the WOAH ring trial (Appendix A).

### 3.4. Diagnostic Specificity

EDTA blood samples from 30 Spanish horses obtained between 2019 and 2022 (AHS-free status area) were negative for all serotypes using triplex TS rRT-PCR, as well as GS rRT-PCR. Additionally, all 11 GS rRT-PCR negative samples from the WOAH ring trial were negative for all nine AHSV serotypes when analysed by triplex TS rRT-PCRs (Appendix A).

### 3.5. Intra-Assay and Intra-Laboratory Repeatability

AHSV reference strains were analysed in ten-fold dilutions (−1 to −7). Ct average and standard deviation were calculated for each sample analysed in duplicate in two different assays (*n* = 4). Since samples were prepared as ten-fold dilutions, different levels of positivity were represented for all nine serotypes. Ct values had optimal repeatability since in both triplex and single assays, the standard deviation was <1 in 90% of the samples and <3 in all the samples.

## 4. Discussion

Typing assays contribute to the eradication of AHSV from defined populations by aiding the choice of current available serotype-specific vaccines. Segment 2, which determines the identity of the virus serotype, is the least conserved of the AHSV segments and presents higher variability among the serotypes. However, hybridisation studies and subsequent nucleic acid sequences of cloned full-length genome segment 2 of the nine AHSV serotypes demonstrated that segment 2 has a sufficient genome conservation level across serotypes to perform serotyping assays by molecular methods [31]. Despite the relevance of the disease, this approach has been explored by a limited number of authors. As a matter of fact, this is the typing strategy currently used by WOAH reference laboratories and other official laboratories involved in AHS diagnosis and control [18] and the same strategy is employed in the typing of other orbiviruses, such as bluetongue [32,33,34,35] or Epizootic hemorrhagic disease [36,37].

Koekemoer, in 2008 [23], developed an AHSV TS RT-qPCR assay that uses a sensor probe and an anchor probe on two different channels, followed by melt curve analysis to differentiate the various AHSV types. This assay showed considerable difficulty in detecting strains of more recent circulation. Additionally, there are two other publications [24,25] describing rRT-PCRs for AHSV typing. However, considering that segment 2 is the most variable segment of the AHSV genome and published assays have not been fully validated, it is highly desirable that more methods are made available to test new emerging AHSV strains.

The standard curves were used to assess the performance of the TS rRT-PCR triplex assays by estimating their efficiency, correlation coefficient, and limit of detection as well as their dynamic range. These curves were calculated using at least six points obtaining R^2^ values over 98%, which indicates high predictive values. The efficiencies were higher than 80% in all serotypes, and >90% in five of them, confirming the good performance of the triplex TS rRT-PCR assays, and a wide dynamic range of at least six dilutions.

Triplex assays targeted to segment 2 have proven to be slightly more sensitive than the GS rRT-PCR (Agüero, 2008) targeted to segment 7 of the AHSV genome, recommended in the WOAH Manual, allowing the typing of any AHSV positive sample. In this regard, as single assays are slightly more sensitive than triplex assays for some serotypes, therefore, the use of single TS rRT-PCRs would be highly advisable in very weak positive GS rRT-PCR samples that turn out negative in the three triplex assays.

Regarding analytical specificity (exclusivity), it should be noted that when AHSV has been grown in Vero cells, a serotype 8 false positive result may occur in TS rRT-PCR triplex 1-6-8 assay. This non-specific result does not occur when serotype 8 is analysed in a TS rRT-PCR single assay. This may be due to a certain combination of primers/probes in the 1-6-8 triplex rRT-PCR that hybridises non-specifically to the Vero cell genome, generating an amplicon that is detected by the serotype 8-specific probe. This fact does not interfere with the diagnosis of the disease in animal clinical samples but must be considered when testing viral suspensions grown in the Vero cell line. In this case, a serotype 8 positive result in the rRT-PCR triplex assay should be confirmed in the serotype 8 TS single assay.

The nine viral suspensions of the AHSV original reference strains (serotypes 1–9) were suspected to contain a mix of several serotypes because of previous VNT results, as previously suggested by other authors [14]. These strains, available at the LCV and other WOAH reference laboratories for AHS, were isolated many years ago from animals in endemic areas, where several serotypes were circulating and therefore co-infecting animals [38]. TS rRT-PCRs have allowed us to confirm the aforementioned hypothesis and have verified that viral suspensions obtained after a process of cloning by limiting dilution contain only the expected serotype. The same process but including specific antisera was applied to the AHS commercial vaccine from Ondestepoort (vials 1 and 2) to obtain viral suspensions containing individual serotypes. These results reinforce the exclusivity of TS rRT-PCRs either in triplex or single assays.

Diagnostic sensitivity has been evaluated employing viral suspensions (*n* = 27) and clinical samples (*n* = 240) from different epidemiological scenarios, allowing the typing of all the positive samples analysed except for one of the weak positive samples included in the WOAH ring trial. However, serotype 4 positive samples are over-represented in the clinical samples, so it would be advisable to include more clinical samples corresponding to other AHSV serotypes from outbreaks in African countries when available. Among the samples included in the assessment of diagnostic sensitivity, there were field samples from outbreaks in AHS-free status countries: Spain (1987–1990) and Thailand (2020).

Spanish outbreaks were caused by the AHSV serotype 4, which was identified by neutralisation methods. Throughout the year 1987, in the central area of Spain, 146 equidae died or were slaughtered, and more than 38,000 animals received a dose of an attenuated polyvalent vaccine which included eight serotypes [2,39]. One year later, the disease re-emerged in Southern Spain (Andalusia) causing the death of horses between the years 1988 and 1990. In the second period of the outbreak (1989–1990), vaccination was performed using an attenuated monovalent vaccine against AHSV4 instead of the previously used polyvalent vaccine [2,39]. The last positive case in Spain was declared in November 1990. The typing results by TS rRT-PCR from Spanish outbreak samples are compatible with the chronology of the events, since in some of the samples corresponding to 1987–1988, when animals were vaccinated with the polyvalent LAV, additional serotypes to AHSV-4 were detected, while in samples corresponding to the period 1989–1990, in which the monovalent LAV vaccine was used, only serotype 4 was detected. Virus isolates SPA1987 serotype 3 (LCV) and SPA1987 serotype 2 (LCV) in Appendix A were isolated from clinical samples 1775/87 and 1903/87 in Appendix A. Typing results from samples and isolates are in agreement.

On 24 February 2020, the first clinical symptoms in horses were reported in Nakhon Ratchasima province (Thailand). In March 2020, the Pirbright Institute (TPI), United Kingdom, as the WOAH reference laboratory for AHS, received EDTA blood samples collected by a private veterinarian and confirmed the presence of AHSV by GS rRT-PCR. On 27 March 2020, the WOAH announced Thailand’s AHS outbreak. Later, TPI reported serotype 1 of AHSV, using the rRT-PCR method described by Bachanek-Bankowska et al., 2014, and Weyer et al., 2015, and subsequently confirmed it by full genome sequencing on virus isolated in KC cells [4,5]. Emergency vaccination with a trivalent vaccine from Onderstepoort including serotypes 1, 3, and 4 (vial 1) started in mid-April, and by mid-September 2020, 9.880 horses had been vaccinated in a single round of vaccination [40]. In this framework, the LCV, as the WOAH reference laboratory for AHS, received the samples officially obtained to confirm AHSV and typing on 14 May 2020, after starting the vaccination campaign. The GS rRT-PCR (Agüero 2008) and triplex TS rRT-PCRs results described in this paper evidenced the detection of serotype 3 in addition to serotype 1 in one of the samples. These results were compatible with the presence of field (serotype 1) and vaccine (serotype 3) viruses in one sample, which could occur given the epidemiological situation at that time. Both in the outbreak of Spain and Thailand, triplex TS rRT-PCR allowed for the detection of field and vaccine strains circulating at the same moment. It would be interesting to carry out subsequent full genome sequencing studies on these samples to analyse the presence of reassortant strains.

In December 2022, an AHS outbreak was notified in Nigeria [41,42], a country where AHS is endemic. Ten clinical samples were received by the LCV, and AHSV serotype 9 was detected in all of them by TS rRT-PCR typing assays. These results were confirmed by sequencing in the LCV.

To complete the assessment of diagnostic sensitivity, GS rRT-PCR positive samples of unknown serotypes from Kenia (2015–2017) and a WOAH Ring trial (*n* = 102) were analysed by TS rRT-PCR triplex assays. It was possible to assign a serotype to all the samples, except one from the WOAH ring trial which had a Ct value of 31.1 in the GS rRT-PCR, so it remains undetermined whether the non-detection in the TS triplex assays could have been due to lack of hybridisation or a small amount of viral RNA in the sample. In all samples, the tests had a positive result for a single serotype and negative results for the rest, even when considering that the samples came from countries with endemic situations, and consequently, the detection of more than one serotype could be expected. Virus isolates KEN 2015 serotype 2 (CVR), KEN 2016 serotype 4 (CVR), and KEN 2017 serotype 5 (CVR) in Appendix A were isolated from clinical samples 2782/15 3, 841/16 8, and 1083/17 3 in Appendix A. Typing results from samples and isolates were in agreement.

TS rRT-PCRs evaluated in this study were routinely used in our laboratory to analyse clinical samples from two experimental infections of vaccinated and unvaccinated horses [30]. They allowed us to unequivocally determine the serotype from both EDTA blood and tissue samples. The AHSV strain used in the challenge (serotype 9), as well as the AHSV strains included in the LAV, were identified by TS rRT-PCRs, and the results were widely discussed.

In summary, our results showed that GS rRT-PCR together with the TS rRT-PCR methods described in this study provide an effective strategy for AHS surveillance and control, especially when multiple serotypes of field or vaccine AHSV strains are involved. In addition, we demonstrate that TS rRT-PCR is relevant for the generation of high-quality reference materials in which the presence of a single serotype is guaranteed. With this aim, the TS rRT-PCRs methods described in this manuscript have been extensively evaluated, placing special emphasis on those parameters related to the suitability of these assays for AHSV typing.

## Figures and Tables

**Figure 1 viruses-16-00470-f001:**
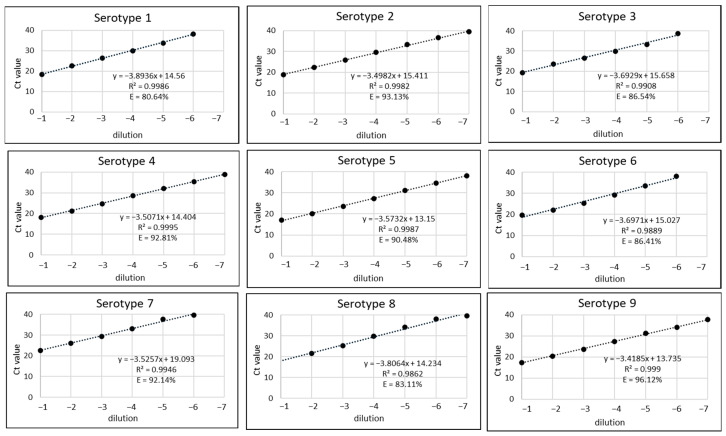
Standard curves for each AHSV serotype (1 to 9) in the TS rRT-PCR triplex assays. E: efficiency; R^2^: correlation coefficient.

**Table 1 viruses-16-00470-t001:** Serotype-specific rRT-PCR primers and probes targeting VP2 AHSV genes for serotypes 1–9. F: Forward, R: Reverse, P: Probe.

AHSV	Primers (F/R) and Probe	Sequence 5′-3′
Serotype 1	AHS-1F	GCAAGCGCTGGCACTTG
	AHS-1R	TTCGAACTCATTCCTTACATCAACA
	AHS1P	FAM-AATGTCTTAGATCGTCAACT-MGB
Serotype 2	AHS-2F	CGGAAACYATGTATTGCCAA
	AHS-2R	TTGTCRTCCTGATCAACCCTAA
	AHS-2P	Cy5-TGAAGGTGCTTACCCGATCTTTCCACA-BBQ
Serotype 3	AHS-3F	AATTATTACAGCGGARAATGCAGTT
	AHS-3R	TCGCACCCCACTCATAACC
	AHS-3P	FAM-AGAGTTGAGGTTGCGGGA-MGB
Serotype 4	AHS-4F	TGAGGTGGAACACGAYATGTC
	AHS-4R	GATATGCCCCCTCACAYCTGA
	AHS-4P	VIC-TATCGGRATTTATGTACAATGAG-MGB
Serotype 5	AHS-5F	GAAGAGACAGGCGATTCAAATGA
	AHS-5R	AAAGCCACCCTTTTTGGTACAAA
	AHS-5P	NED-TGTTGARATGCTGAGGC-MGB
Serotype 6	AHS-6F	AGCCAGGGCTTCTTTGCA
	AHS-6R	CTCATGTTCAACCCACTGTACATTAA
	AHS-6P	VIC-GTCATCACCGTAAGCG-MGB
Serotype 7	AHS-7F	GAGTGGGGCGCAACAAATC
	AHS-7R	CCGGATATGCACCCTCACAT
	AHS-7P	VIC-CGATATCTGAGTTCATGTATGAA-MGB
Serotype 8	AHS-8F	GAAATTATCAGCGGACTGACTAAGAA
	AHS-8R	AAACATCTACCTTTTGCGAATCTTG
	AHS-8P	NED-ACGTGATTCTTTTCCC-MGB
Serotype 9	AHS-9F	TACTGTGTCGGTGAGGGATTTT
	AHS-9R	GGCACGACCGGATATGA
	AHS-9P	FAM-TTCACATTTCGTTTGTTT-MGB

**Table 2 viruses-16-00470-t002:** TS rRT-PCR results obtained from AHSV reference strains and AHSV OBP vaccine (vials 1 and 2) before and after virus cloning by limiting dilution, to evaluate the exclusivity among AHSV serotypes.

Strain	Positive Serotypes in Triplex TS-rRT-PCRs Assays (Ct Value) *		Specific Antisera Used in the Cloning Process
	Before Cloning	After Cloning	
AHSV1 29/62 reference strain	1 (14.7) and 4 (26.4)	1 (18.3)	without antiserum
AHSV2 82/61 reference strain	2 (15.2)	2 (18.9)	without antiserum
AHSV3 13/63 reference strain	1 (24.8) and 3 (15.0)	3 (19.4)	without antiserum
AHSV4 32/62 reference strain	4 (13.5)	4 (18.1)	without antiserum
AHSV5 30–62 reference strain	3 (15.6) and 5 (18.9)	5 (17.2)	without antiserum
AHSV6 39/62 reference strain	6 (19.8) and 7 (21.3)	6 (19.7)	without antiserum
AHSV7 62/31 reference strain	3 (15.5) and 7 (18.3)	7 (22.7)	without antiserum
AHSV8 62/10 reference strain	5 (28.2) and 8 (15.5)	8 (22.1)	without antiserum
AHSV9 90/61 reference strain	9 (15.0)	9 (17.4)	without antiserum
AHSV commercial vaccine OBP vial 1 (serotypes 1, 3 and 4)	1 (17.2), 3 (15.5) and 4 (17.9)	1 (14.0)	serotypes 3 and 4
		3 (13.9)	serotypes 1 and 4
		4 (14.7)	serotypes 1 and 3
AHSV commercial vaccine OBP vial 2 (serotypes 2, 6, 7 and 8)	2 (22.0), 6 (21.8), 7 (23.9) and 8 (21.7)	2 (17.4)	serotypes 6, 7 and 8
		6 (18.4)	serotypes 2, 7 and 8
		7 (17.7)	serotypes 2, 6 and 8
		8 (13.6)	serotypes 2, 6 and 7

OBP: Onderstepoort Biological Products. * The remainder serotypes were negative.

## Data Availability

The data that support the findings of this study are available from the corresponding author upon reasonable request.

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
