# Peer review of "Development and Validation of Three Triplex Real-Time RT-PCR Assays for Typing African Horse Sickness Virus: Utility for Disease Control and Other Laboratory Applications"

_viruses, 2024, doi:10.3390/v16030470_

Round 1

Reviewer 1 Report

Comments and Suggestions for Authors

The article entitled “Development and validation of three triplex real time RT-PCR for typing African horse sickness virus: utility for disease control and other laboratory applications” by Villalba et al. describes the development and validation of three triplex real time RT-PCR (rRT-PCR) methods to quickly determine the AHSV serotype. This paper describes the development, optimization and evaluation of a set of three triplex rRT-PCR assays which provide rapid and reliable AHSV serotype identification. Serotypes 2, 4 and 9 were included in the first triplex assay, serotypes 3, 5 and 7 in the second and serotypes 1, 6, and 8 in the third triplex rRT-PCR. They allow discrimination of mixed infections and have been used to identify incursions of multiple AHSV types in endemic and non-endemic situations. The methodology developed in this work is well described and the results adequately analyzed. The three triplex real time RT-PCR described in this article will provide an effective strategy for AHS surveillance and control, especially when several serotype of field or vaccine AHSV strains are involved.

Author Response

Dear Reviewer 1,

Thank you for your comments. We hope that the methods described in this work can be useful for controlling the disease.

Reviewer 2 Report

Comments and Suggestions for Authors

The paper entitled “Development and validation of three triplex real time RT-PCR for typing African horse sickness virus: utility for disease control and other laboratory applications” defines a new diagnosis methodology for AHSV. The aim of the study is well established and results are clear. However, I would like to make a few remarks and some recommendations to improve the manuscript.

Please use English (s) or USA form, but the same in all cases. Line 384: neutralization. Line 84: neutralisation tests (VNT) and seroneutralisation

Line 86: cross-reactive. Explain that cross-reaction is only between some serotypes

Line 92: when you say “availability”, do you mean the combined use of several rRT-PCR methods. Please clarify this sentence.

Table 1: add to the legend. F: Forward. R: Reverse. P: Probe

Please change 1-8 serotypes of EHDV (in Lines 109, 190 and 251) by serotypes of EHDV (EHDV-1, -2, -4, -5, -6, -7, and -8) or (1-2, and 4-8). Otherwise, the reader will think that 8 serotypes have been officially assigned instead of seven.

Improve quality (resolution of graphs)

Author Response

Dear Reviewer 2,

Thank you for your comments and suggestion. We agree to implement them and believe that it has improved the manuscript.

Please use English (s) or USA form, but the same in all cases. Line 384: neutralization. Line 84: neutralisation tests (VNT) and seroneutralisation. It has been reviewed

Line 86: cross-reactive. Explain that cross-reaction is only between some serotypes. There were explained which cross-reactions btween serotypes are described in bibliography.

Line 92: when you say “availability”, do you mean the combined use of several rRT-PCR methods. Please clarify this sentence. The sentence has been modified to clarify it.

Table 1: add to the legend. F: Forward. R: Reverse. P: Probe. Done.

Please change 1-8 serotypes of EHDV (in Lines 109, 190 and 251) by serotypes of EHDV (EHDV-1, -2, -4, -5, -6, -7, and -8) or (1-2, and 4-8). Otherwise, the reader will think that 8 serotypes have been officially assigned instead of seven. Done.

Improve quality (resolution of graphs). Figure 1 has been included uploaded in its editable version improving resolution.. 

Reviewer 3 Report

Comments and Suggestions for Authors

This study is well presented and the conclusions are supported by the data. However, there are many areas of the text that require improvement. While these are generally minor edits, they are frequent. Some sections of the text could be shortened to reduce repetition - for example, lines 116-124, 135-144.

While the authors have presented all of the data perhaps just a selection of the text could be included in the main manuscript and the remainder included as supplementary data. Tables 4-8 could be transferred and in particular, very long tables such as Table 8.   

Comments on the Quality of English Language

See previous comments. In particular, attention should be given to the use of singular and plural (eg line 32- degree). Other examples: L41 'affecting to'; L47 'named to'; L105'limit dilution' should be 'limiting dilution' throughout text; L131 'centrifugation 10 minutes'L331' get serotyping; L354 'highly recommendable' L359 'does not happen' L398 & 433 - what is meant by 'in correspondence' 

Author Response

Dear Reviewer 3,

Thank you for your comments and suggestions. We agree with the suggestion to include several tables in the supplementary material. English language was reviewed by an expert to improve it.

There are many areas of the text that require improvement. While these are generally minor edits, they are frequent. Some sections of the text could be shortened to reduce repetition - for example, lines 116-124, 135-144. The text has been extensively revised and several paragraphs have been redrafted.

While the authors have presented all of the data perhaps just a selection of the text could be included in the main manuscript and the remainder included as supplementary data. Tables 4-8 could be transferred and in particular, very long tables such as Table 8.   Tables 4 to 9 have been included in supplementary material as well as Table 2 because relevant information of this table is summaried in figure 1.

Comments on the Quality of English Language

See previous comments. In particular, attention should be given to the use of singular and plural (eg line 32- degree). Other examples: L41 'affecting to'; L47 'named to'; L105'limit dilution' should be 'limiting dilution' throughout text; L131 'centrifugation 10 minutes'L331' get serotyping; L354 'highly recommendable' L359 'does not happen' L398 & 433 - what is meant by 'in correspondence'. The English language has been reviewed by an expert and many modifications have been made to improve the text.